## REVIEW ARTICLE

# Mapping and comparing fMRI connectivity networks across species

Marco Pagani [1,2,3], Daniel Gutierrez-Barragan [1], A. Elizabeth de Guzman[1],
Ting Xu [4] & Alessandro Gozzi [1✉]

Technical advances in neuroimaging, notably in fMRI, have allowed distributed patterns of functional connectivity to be mapped in the human brain with increasing spatiotemporal resolution. Recent years have seen a growing interest in extending this approach to rodents and non-human primates to understand the mechanism of fMRI connectivity and complement human investigations of the functional connectome. Here, we discuss current challenges and opportunities of fMRI connectivity mapping across species. We underscore the critical importance of physiologically decoding neuroimaging measures of brain (dys)connectivity via multiscale mechanistic investigations in animals. We next highlight a set of general principles governing the organization of mammalian connectivity networks across species. These include the presence of evolutionarily conserved network systems, a dominant cortical axis of functional connectivity, and a common repertoire of topographically conserved fMRI spatiotemporal modes. We finally describe emerging approaches allowing comparisons and extrapolations of fMRI connectivity findings across species. As neuroscientists gain access to increasingly sophisticated perturbational, computational and recording tools, cross-species fMRI offers novel opportunities to investigate the large-scale organization of the mammalian brain in health and disease.

**Mapping functional connectivity networks with fMRI**. A large body of evidence suggests that, even at rest (i.e. without any overt cognitive or sensory stimulation) brain activity synchronously fluctuates in concert across distributed regions[1–3]. This important observation has promoted an emerging model whereby brain activity and complex cognitive functions can be conveniently operationalized as the result of a flexible and dynamic interaction between nodes that constitute distributed functional networks[4,5].

A popular approach for studying how brain regions are intrinsically organized and communicate with each other is resting state fMRI (rsfMRI). Initial observations that rsfMRI signal is highly synchronous between functionally related regions[6] have promoted a framework whereby statistical dependencies between spontaneous fluctuations in brain activity are interpreted as an index of interareal communication, or "functional connectivity" between regions[7]. Subsequent investigations have revealed reproducible patterns extending beyond motor sensory areas to encompass other sensory systems and distributed networks of regions that are highly synchronous at rest, and whose activity is modulated by cognitive tasks[8]. These spatiotemporal patterns of coordinated activity define a set of reproducible resting-state networks (RSNs) characterized by high intrinsic functional connectivity[9]. Importantly, RSNs topography has also been shown to spatially reconstitute several known functional systems of the

[1] Functional Neuroimaging Laboratory, Center for Neuroscience and Cognitive Systems, Istituto Italiano di Tecnologia, Rovereto, Italy. [2] Autism Center, Child Mind Institute, New York, NY, USA. [3] IMT School for Advanced Studies, Lucca, Italy. [4] Center for the Integrative Developmental Neuroscience, Child Mind Institute, New York, NY, USA. ✉email: alessandro.gozzi@iit.it

human brain, thus suggesting that intrinsic and task-evoked brain activity may strongly influence each other[1,8,10].

The accessibility and practicality of rsfMRI, along with recent progress in open data initiatives and the development of advanced analytical methods have greatly improved our understanding of the intrinsic functional organization of the human brain[11,12]. Prominent investigations of rsfMRI networks have characterized how patterns of intrinsic brain activity and functional connectivity are related to cognitive function and behavior[10,13]. Similarly, rsfMRI mapping in pathological states has revealed atypical network activity and altered functional connectivity in all major neurological and psychiatric diseases[14,15].

While many ongoing brain mapping initiatives heavily rely on the use of rsfMRI, the correlative nature of this approach and our inability to interpret and physiologically decode the mechanisms underlying interareal fMRI coupling strongly limit the impact of this research, often relegating it to phenomenological evidence devoid of any mechanistic information. As a result, rsfMRI connectivity mapping is still considered by many as an imaging endophenotype, or proxy for interregional communication of uncertain physiological significance.

**Bridging the explanatory gap: mechanistic studies of fMRI connectivity.** Many key questions related to the fundamental principles of rsfMRI connectivity remain to date unaddressed, primarily as a consequence of our inability to mechanistically disambiguate a highly correlative phenomenon like rsfMRI in humans. While this issue has not prevented a proficient use of rsfMRI connectivity in cognitive neuroscience, a deeper understanding of the physiological basis of rsfMRI may equally benefit cognitive neuroscientists and researchers interested in the organization of functional networks in brain disorders. A better understanding of the mechanism of rsfMRI may for example prevent the erroneous (yet common) use of this metric as a monotonic index of connectivity[16], or may help understand its relationship with internal states and physiological signals that bias network organization e.g. arousal and peripheral signals[17]. Furthermore, research into the physiological bases of rsfMRI coupling may help decode patterns of dysconnectivity in brain disorders and relate them to physiologically interpretable measures that could help diagnose or stratify patient populations for customized treatments.

More broadly, mechanistic investigations of fMRI connectivity can help bridge the big explanatory gap that exists between molecular and biophysical modeling of microscale neural activity, and network-level descriptions of brain function (Fig. 1a)[18]. Human studies have attempted to address this issue by relating patterns of fMRI activity to measures of anatomical connectivity or physiological measurements recorded concurrently with rsfMRI (reviewed by Suárez et al.[4]). Similarly, large-scale imaging initiatives have linked genetic variation to network features both in healthy populations[19,20] as well as in carriers of genetic alterations associated with brain disorders[21–24]. However, the correlative nature of these investigations has not allowed a reliable mechanistic disambiguation of fMRI (dys)connectivity.

Attempts to establish causality using brain stimulation, or by relating patterns of altered rsfMRI connectivity to neurological lesions have revealed interesting associations between clinical scores and rsfMRI network dysfunction in humans, hence substantiating the role of distributed network activity in mediating complex behavioral traits and functions[25–29]. However, our poor comprehension of the complex physiological cascade underlying neurological damage and the similarly limited understanding of the physiological basis of brain stimulation limit the mechanistic significance of these investigations. Crucially, the

recent extension of rsfMRI mapping to physiologically accessible species offers the unprecedented opportunity to bridge the "explanatory gap" (Fig. 1a). Specifically, the combination of rsfMRI with the ever-increasing set of advanced neuromanipulations and recording tools available in animals provide a means to causally deconstruct rsfMRI activity, relate central and peripheral physiological events to brain-wide patterns of fMRI activity, and investigate the anatomical correlates of brainwide network activity across investigational scales (Fig. 1b).

This experimental approach is coming of age[30], and promises to add novel interpretative dimensions to rsfMRI and its use in cognitive and translational neuroscience. The impact and potential of mechanistic investigation in animals is epitomized by a series of investigations that we briefly summarize in the following sections, with the intent to make readers aware of the critical contribution and impact preclinical fMRI is having in the field. Crucially, these examples also highlight the need to identify reliable means to map and relate fMRI connectivity networks (and related findings) across species to ensure an efficacious transfer of information from and to human and animal studies.

**Mechanistic studies in non-human primates.** fMRI connectivity mapping was initially extended from human to non-human primates (NHP), where its combined use with invasive electrophysiological recordings and stimulations has produced a foundational understanding of the neural correlates of blood oxygenation level dependent (BOLD) fMRI activity[31–38]. Further electrophysiological studies have linked rsfMRI activity to resting local field potentials[39], neural coupling[40], and physiological signatures of arousal[41].

fMRI studies in NHP have also benefitted from the availability of detailed anatomical information attainable via use of invasive axonal tracers[42–44]. By relating topographical patterns of fMRI connectivity to the organization of the NHP axonal connectome, NHP studies have supported the notion that fMRI network organization is strongly constrained by underlying patterns of anatomical connectivity[37,45,46]. The generalization of this notion has been further investigated in interventional studies encompassing the use of lesions[46,47] or reversible inactivation of brain activity via infusion of drugs[48] and chemogenetics ligands in virally-transfected animals[49].

Further interventional studies in NHP have encompassed the use of electrical, ultrasound[50,51], or infrared stimulation[52] to understand how the activity of a single brain region causally affects global patterns of fMRI connectivity. The use of sedative and anesthetic agents has also been explored to map and compare patterns of fMRI activity as a function of brain state. For example, a study comparing wakeful versus unconscious states has shown that spontaneous functional connectivity patterns in awake monkeys show a rich repertoire of functional connections that is more dissimilar to structural connectivity compared to anesthesia[53,54]. Differences in cross-subjects variability of fMRI connectivity patterns has been described in the cortex of awake vs. anesthetized NHP[55]. Cortical BOLD responses to visual stimulation are instead consistent in awake vs. unconscious NHP[56]. Finally, it has also been shown that the intrinsic network structure of main primary and associative networks in the anesthetized macaque monkeys are topographically similar to those mapped in conscious humans[37].

Finally, attempts to map fMRI connectivity in transgenic primate models of human disorders have also been recently described[57].

The impact of NHP investigations of the neural correlates of BOLD signal and fMRI connectivity has been substantial. As novel recording and manipulation tools become available to NPH

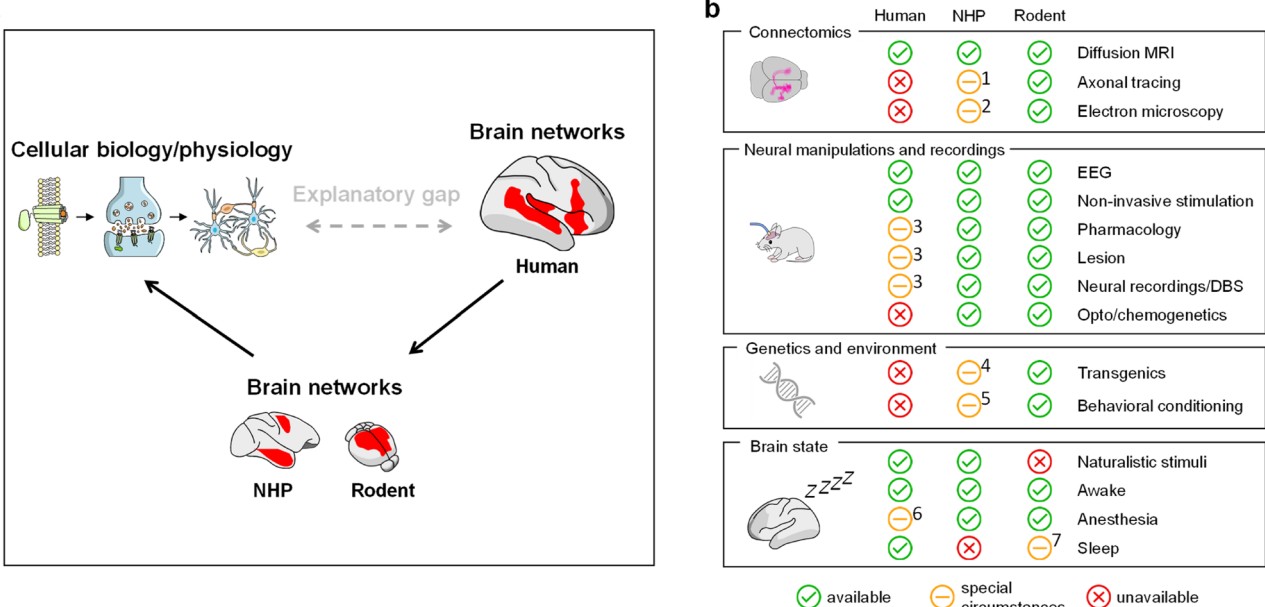

**Fig. 1 Bridging the explanatory gap. a** A major gap exists between models of human brain function at different levels of inquiry. The use of neural systems measurements like fMRI in physiologically accessible animals can help bridge this gap by causally linking neural events at different investigational scales. Another major advantage of this approach is related to the possibility of probing the role of specific microscale processes in shaping whole brain patterns of fMRI connectivity. **b** Highly complementary experimental approaches can be implemented in humans, non-human primates (NHPs) and rodents to investigate the neural basis and organization of fMRI connectivity. The panel illustrates, for each species, experimental approaches or resources that are readily available and commonly used (green), currently unavailable or ethically inaccessible (red), and available under special circumstances (yellow), or for which only proof of concept studies have been described. For the latter group, references to recently published examples are reported here: [1][43,135,222], [2][223], [3][27], [4][224], [5][225], [6][205] [7][226]. Non-invasive stimulation includes transcranial magnetic stimulation (TMS), transcranial alternating current stimulation (tACS) and related methods. EEG: electroencephalography; DBS: deep brain stimulation.

researchers, the added value of mechanistic investigation in these species is expected to exponentially grow, offering novel opportunities to bridge theoretical and experimental neuroscience, and reliably translate findings from and to human research. However, despite its tremendous potential, the implementation of fMRI in NHPs remains technically, logistically and procedurally very demanding, requiring highly specialized equipment and personnel, and high investment and running costs. Of equal importance, ethical constraints prevent a full exploitation of the ever-expanding repertoire of manipulations and recording methods available to experimental neuroscientists.

**Mechanistic studies in rodents.** To circumvent these limitations, recent years have seen a strong drive towards the implementation of fMRI connectivity mapping in more physiologically accessible species, such as rats and mice. The field of rodent fMRI has steadily grown, and now encompasses a large number of laboratories in the world[58–61]. Much of the interest in this approach lies in the possibility of combining fMRI connectivity mapping with recordings and experimental manipulations (including genetically modified lines) not directly applicable for technical or ethical reasons in higher mammalian species.

The possibility of combining fMRI connectivity mapping with cell-type specific neuromanipulations represents a very promising area of investigation in the field of rodent fMRI. Notable examples of this approach have investigated the contribution of ascending transmitter nuclei (e.g. noradrenergic, cholinergic, serotonergic or dopaminergic neurotransmission) in causally shaping the organization of the fMRI connectome via chemogenetic[62–64] or optogenetic manipulations[65,66]. Further studies have employed chemogenetic manipulations to physiologically-decode fMRI connectivity signals[67] and study how activity in one region causally affects whole-brain patterns of

fMRI connectivity[68]. By linking neural signals to large-scale patterns of fMRI connectivity these studies are uncovering a set of fundamental principles governing communication between brain regions at the macroscale, and in some cases challenging prevailing interpretations of fMRI connectivity[16].

The wide availability of transgenic models in rodents represents an important complement to mechanistic investigations in NHP, where the use of transgenes is still limited (Fig. 1b). A fertile area of application of fMRI in transgenic models is to investigate how individual disease-risk genetic mutations affect brain wiring and fMRI connectivity. For example, this approach has been used by us and others to link autism-associated risk genes to specific macroscale signatures of fMRI dysconnectivity, thus corroborating a key role of genetics in shaping the functional connectome in developmental disorders[69–76]. By mechanistically relating dysfunctional fMRI activity to atypical axonal wiring[70,72,74,77], synaptic abnormalities[71] or electrophysiological alterations[74], some of these studies have shed light on the neural mechanisms underlying connectional atypicality. Multisite extensions of this approach have revealed the effect of genetic variability across multiple genetic models of autism[78]. Notable applications of fMRI in rodents have also been extended to other neurological and psychiatric disorders (reviewed in[79]), to include Alzheimer's disease[80,81], Huntington disease[82], schizophrenia[83] and ADHD[84].

Rodent fMRI studies have also leveraged the availability in the mouse of a high-resolution, directed, mesoscale axonal connectome[85] to uncover the relationship between functional and anatomical connectivity. This resource has made it possible to extend investigations of the structural basis of network activity to finer spatial scales inaccessible to human investigators[86,87], as well as to probe axonal correlates of sub-networks of the rodent brain, such as the mouse default mode network (DMN)[88] or salience network (SN)[89].

Recent years have also seen great advancements in linking macroscale fMRI mapping with mesoscale neural signals, particularly through the use of genetically encoded calcium indicators via fiber photometry. This innovative approach enabled the prediction of MRI network organization based on patterns of calcium neural activity in specific brain regions and the establishing of reciprocal causal interactions between networks systems of translational relevance via calcium-related signals[90–93]. More importantly, simultaneous mapping of whole brain fMRI and cortex-wide mesoscale calcium imaging has been recently demonstrated, offering novel opportunities to relate neural and hemodynamic signals across multiple investigational scales[94].

We conclude this overview by mentioning the widespread use, both in NPH and rodent fMRI studies, of intravascular contrast agents aimed at increasing functional contrast to noise ratio[95], reduce susceptibility artefacts, and in turn increase the sensitivity of both evoked and intrinsic functional imaging[96–98]. Superparamagnetic iron-oxide nanoparticles in particular have been widely used to map cerebral blood volume[99], producing vascular contrast that reproduces the macroscale organization of conventional BOLD fMRI connectivity networks[96,100]. The high vascular specificity of this approach and its practicality makes it a robust approach for mapping fMRI networks in small mammals like rodents and NHP.

Collectively, complementary fMRI research in NHP and rodents offers the opportunity to mechanistically probe the neural basis of fMRI coupling in the mammalian brain. However, to best leverage and properly contextualize the wealth of information generated by animal fMRI studies, methods to extrapolate findings across species are required. At the core of this argument lies the critical need to understand whether the mechanistic inferences made in lower mammalian species, especially those pertaining to specific networks or circuits, are species-invariant (or not), and if so, to which extent they can be topographically related to network systems of the human brain. Addressing this question is of special importance for the investigation of mechanisms that influence fMRI network topography such as the contribution of central modulatory systems, or mechanism of dysconnectivity produced by genetic or developmental insult[30].

## Organization of fMRI networks across species

**Species-invariant and species-specific fMRI connectivity networks**. While the question of how to reliably match network organization across-species remains open, correspondences in the organization of fMRI network across the phylogenetic tree suggest that topographical organization of large-scale networks can be used to index patterns of regional activity and guide the extrapolation of fMRI finding to and from mammalian species. The ultimate goal of this line of investigation is to identify homologous functional networks enabling comparative brain mapping and the extrapolation of region- or network-specific findings across the phylogenetic tree. The cogency of this approach is supported by the results of recent studies revealing a set of evolutionary precursors of human RSNs in NHP and lower mammalian species.

Initial investigations in macaques[37], followed by more recent studies in lower mammalian species (i.e. marmosets, lemur monkeys, rats and mice)[100–103], have revealed that that the organization of spontaneous fMRI signal into distributed and bilateral spatiotemporal patterns is a foundational, evolutionarily-conserved property of the mammalian brain. Further investigations of the rodent and primate cortices have revealed the presence of phylogenetically-conserved RSNs, including

interhemispheric synchronous visual, auditory[104,105] and somatomotor networks[55,88,100,105–108]. The latter has been termed in rodents "latero-cortical network" (LCN) owing to its anterolateral distribution[107]. Owing to their unambiguous location, and their anchoring on well characterized cortical territories (in each species), visual, auditory, and somatomotor networks can thus be considered "species-invariant" networks, i.e. networks that can be reliably identified and mapped across the phylogenetic tree. Clearly, species invariance does not imply here absolute spatial, cytoarchitectural or connectional homogeneity, but rather an unambiguous representation of these systems across multiple species, independent of their evolutionary complexity.

The observation of distributed fMRI networks in the human brain encompassing higher order areas such as the default mode or salience network[8,109] led to the initial assumption that the organization of these systems, could reflect a distinctive organizational mode that is exclusive to humans. However, the discovery of similarly distributed connectivity networks also in anesthetized NHPs has refuted this hypothesis[37,110]. Further studies in NHP and rodents have identified RSNs that anatomically reconstitute the topography of the human DMN and salience networks (SN)[60,88,89,100,110,111]. Interestingly, similar to what is observed in humans[2,60] somatomotor regions of the LCN exhibit anticorrelation with the DMN[55,100,106–108] (Fig. 2a). It should be noted that all these network correspondences hinge on the presence of synchronous activity in phylogenetically conserved regions such as limbic and somatomotor cortical areas, or evolutionarily-ancient precursors of the medial prefrontal cortex like the anterior cingulate cortex[112]. Therefore, in lower species, the organization of these fMRI network precursors only partly recapitulate the richer topography of corresponding systems in humans[88,103,113].

In keeping with this, plausible rodent and NHP precursors of human higher-order cognitive networks such as the dorsal-attention, temporoparietal or executive control networks remain to be firmly identified. Being these systems anchored on high-order cortical areas that have greatly expanded and that are highly specialized in humans with respect to primates and rodents, the possibility that these systems are human-specific seems highly plausible[60,114]. We note however that reports of a cortical system subserving executive control functions in NHP have been published[115–117] (Fig. 2a). Further research into the phylogenetic evolution of these networks is required to address the question of whether and how these functions are represented at the network level in lower species.

Importantly, notable cross-species correspondences in fMRI network organization also encompass subcortical systems. Seed-based mapping or independent component analyses in rodents have allowed the break down of fMRI network activity into fine-grained subcortical functional systems, including hippocampal, basal ganglia, thalamic and basal-forebrain networks[106,118–120]. Analogous systems can be reliably identified both in NHP and humans[121–126]. While the evolutionarily ancient origin of these systems makes them more easily identifiable and directly related across species, subcortical systems may also exhibit increasingly complex topography along the phylogenetics trees similar to what is observed in cortical networks[125], especially in terms of integration and cross-talk with other anatomical substrates. These factors should be accounted for when mapping and comparing activity in subcortical networks systems across species.

**General organizational principles of fMRI connectivity across species**. Irrespective of some expected differences in the organization of higher-order cortical networks along the phylogenetic tree, some generalizable, evolution-invariant principles underlying the functional organization of RSN are apparent, which can guide topographical mapping of networks systems across species.

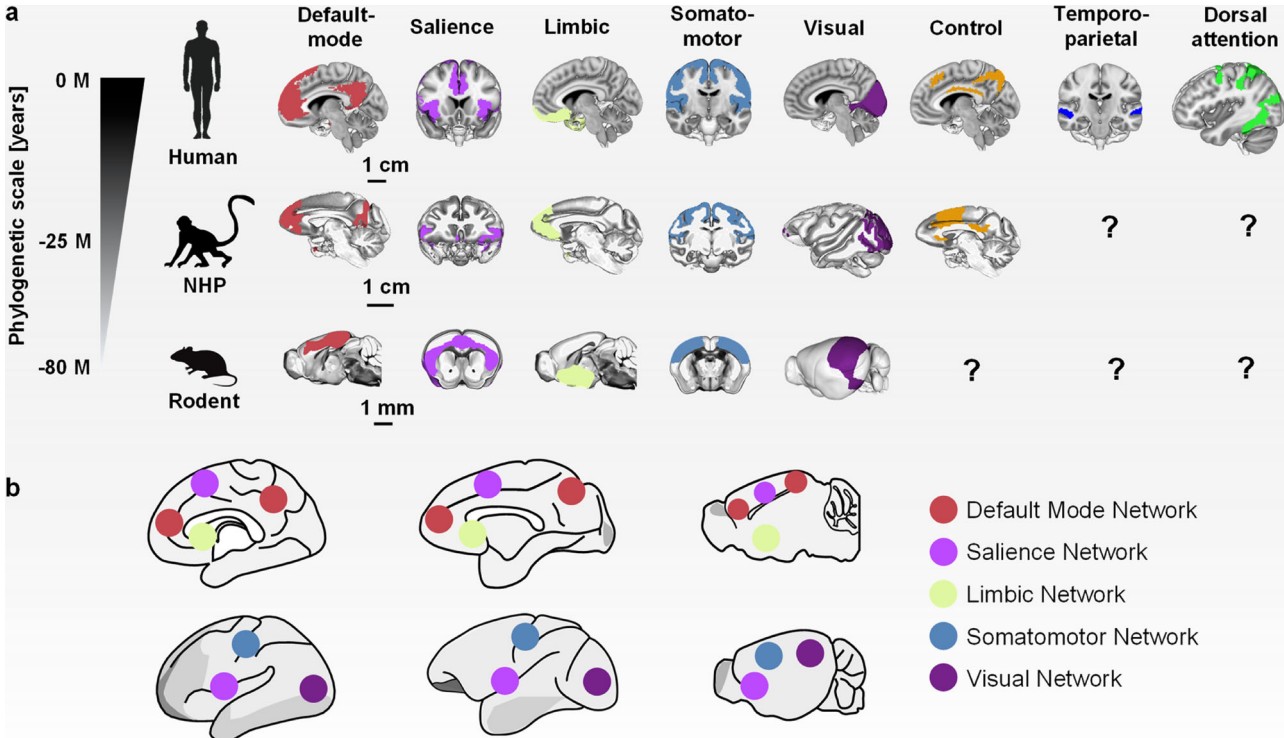

**Fig. 2 Species-invariant and species-specific fMRI connectivity networks. a** Representative cortical fMRI networks in humans[227] and their putative evolutionarily precursors in awake NHP[55] and rodents[106]. Plausible homologs of the human default mode, salience, limbic, somatomotor, and visual networks have been described in NHP[55] and mice[106]. A possible homolog of the executive control network has been identified in NHP[162]. To date, no evolutionarily plausible precursors of human dorsal attention and temporoparietal networks have been identified in animals. **b** Approximate location of hub constituents of default mode, salience, limbic, somato-motor and visual networks in the human, NHP and mouse brain as shown in later (top panel) and sagittal (bottom) panel. Please note that the Yeo parcellation[227] of the human brain incorporates the auditory network within somato-motor areas. Evidence of dissociable auditory networks has been however reported in all species, including mice[100] rats[139] and NHP[117]. Default mode network is in red, salience network is in violet, limbic network is in yellow, somatomotor network is in blue, visual network is in purple, control network is in orange, temporoparietal network is in dark blue, and dorsal attention network is in green, according to Yeo's parcellation[227].

The most conspicuous of these principles is the organization of cortical and subcortical RSN into exquisitely bilateral, homotopic systems exhibiting robust interhemispheric synchronization. This feature applies to virtually all fMRI networks so far identified in lower mammalian species and encompasses associative (i.e., DMN, salience, etc.) as well as sensory and somatomotor (i.e., visual, lateral cortical etc.) networks. Interhemispheric homotopicity is also a defining characteristic of subcortical rsfMRI network[100,126,127].

The presence of a tight relationship between the topography of anatomical and functional connectivity is a second foundational organizational principle of fMRI connectivity across species. Tracer-based high-resolution connectome mapping in rodents and NHPs have expanded the results of initial investigations of the structural basis of fMRI connectivity in humans based on diffusion weighted MRI[128–130], by allowing fMRI network architecture to be related to quantitative and directional measures of connectivity at the mesoscale[85,131,132]. These studies have revealed conserved rules of cortical connectivity across species[133], including cell-class-specific projection patterns[88,134]. Importantly, this relationship appears to be independent of the spatial scale investigated[135]. These correspondences are of interest in the light of an emerging body of evidence supporting the notion that topographic organization of fMRI networks is shaped and constrained by the underlying anatomical organization of the brain, with contributions that go beyond and above axonal connectivity[4,86,87], to also include laminar microstructure of myelination[136,137] and the geometric organization of brain anatomy[138].

The presence of hub-like nodes in associative regions is an additional organizational feature that appears to underpin the global structure of fMRI connectivity in multiple mammalian species. fMRI connectivity hubs have been shown to be similarly anchored to midline components of the DMN in rodents[107,139], NHPs[140] and humans[141,142], albeit with a shift from more anterior (medial prefrontal) to posterior (retrosplenial, posterior cingulate cortex) cortical areas occurring across the evolutionary timeline[60,107] (Fig. 2b). Interestingly, a general overlap between the anatomical location of the functional and structural connectivity hubs has been observed in rodents[86,143,144], NHP[140,145] and humans[129,146]. These correspondences further corroborate the intimate relationship between anatomical and functional organization of the mammalian brain we have described above.

**fMRI connectivity gradients**. The spatial correspondence between structural and fMRI connectivity in cortical areas has been recently investigated in greater depth as part of a broader question of high evolutionary relevance: why is each brain region located where it is, and how does its anatomical organization impact interareal functional communication? This is an old question that has fascinated researchers for many years[147]. Classical investigations of cortical microarchitecture and cortico-cortical connectivity have proposed a hierarchical model whereby regions having similar features occupy the same position along a graded cortical axis[148–150]. According to the "tethering hypothesis", high-order association cortices untethered from sensory

hierarchies in the cortical expansion during evolution, resulting in their current topological location as in-between zones of the cortical mantle[151–153]. Interestingly, recent studies have expanded this framework by showing that multiple biological properties that are relevant to the organization of functional connectivity are also distributed along a primary axis in spatial variation ranging from sensory (i.e. unimodal) to transmodal cortical regions. These include cortical profiles of axonal connectivity[86], gene expression[154], receptor expression[155], as well as cytoarchitectural properties such as local neuronal density[156], dendritic spine density and size and dendritic tree complexity[157], all of which are organized a common axis of spatial variation that is recognizable in humans[158], NHP[136] and rodents[159].

Recently, Margulies and colleagues[160] captured this unimodal to transmodal axis (i.e. gradient) with fMRI using a diffusion embedding approach, thus demonstrating that the topological organization of cortical fMRI connectivity is hierarchically organized along a spatial gradient spanning unimodal-transmodal cortical regions in both humans and NHP (Fig. 3a). This principal gradient is also found to be spatially aligned with T1-T2-weighted inferences of intracortical myelin content, thus relating macroscale RSN organization to classical sensory-transmodal hierarchy and its underlying microstructural foundations[150,158,161]. Similar fMRI-based connectivity gradients have also been recently identified in macaques, marmosets and rodents where they were found to recapitulate the unimodal-transmodal axis mapped in the human cortex (Fig. 3b), and to be exquisitely aligned with the organization of the axonal connectome[86,162–164]. Notably, it has been recently suggested that this gradient is representative of the course of cortical evolution, with phylogenetically newer areas emerging at particular points along the axis[133,165].

Collectively, these lines of evidence suggest that the spatial topology of cortical fMRI connectivity along preordered evolutionarily-relevant gradients is a fundamental organizational principle of mammalian fMRI connectivity across species[86,162,164,166] Above and beyond the use of network-specific inferences, this organizational axis may also provide valuable evolutionarily-relevant landmark coordinates to enable functional alignment of fMRI networks across the phylogenetic tree (Fig. 3).

**fMRI spatiotemporal modes.** Our discussion of fMRI connectivity has so far been based on canonical descriptions of interareal fMRI coupling as a time-invariant measure of static functional connectivity. However, the brain is inherently a dynamical system, and a large body of evidence over the past ten years has demonstrated that the correlation structure of fMRI signal continually evolves over a time scale of seconds[167–171]. fMRI networks dynamics can be mapped using different computational approaches that are referred to under the umbrella term "dynamic functional connectivity" (dFC). A key goal of these approaches is the identification of recurring fMRI network configuration whose occurrence and temporal sequence determine the steady state structure of "static" fMRI connectivity networks[172]. Using dFC based on sliding correlation windows, the dynamic nature of fMRI connectivity networks have been initially identified in humans[167–169], NHPs[173] and later in rodents[174].

While a thorough description of all the contemporary approaches used in dFC is beyond the scope of this review (see[172]), here we focus on a set of recently developed dFC methods, as they have proven effective in revealing evolutionarily relevant principles in the dynamic organization of fMRI networks across species[175] The framework of these methods is designed to describe the intrinsic dynamics of BOLD fMRI signal with voxel-resolution (i.e. without the necessity of a brain parcellation), in non-correlational terms. Theoretical and experimental work

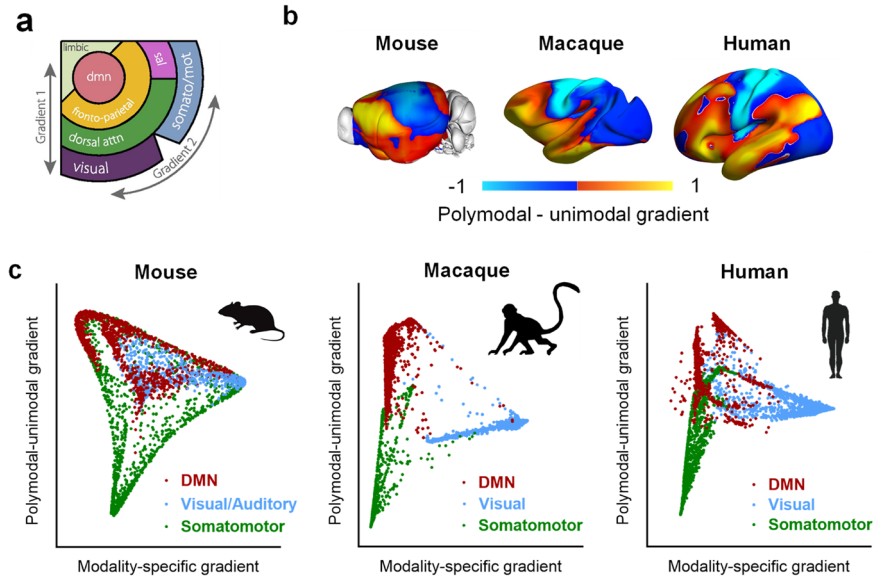

**Fig. 3 The topography of cortical connectivity gradients is phylogenetically conserved. a** fMRI connectivity in the human cortex is arranged along two main axes. The first and dominant gradient spans unimodal-polymodal cortical regions. The second gradient reconstitutes the sensory specific arrangement of cortical regions. Adapted with permission from[160]. **b** The topography of the dominant fMRI gradient is evolutionarily conserved in the mouse[86], macaque monkey[162] and human[162] cortex. Adapted with permission from[86] © The Authors, some rights reserved; exclusive licensee AAAS. Distributed under a CC BY-NC 4.0 License (http://creativecommons.org/licenses/by-nc/4.0/"). Adapted with permission from[162]. **c** Reciprocal organization of unimodal-polymodal and modality-specific gradient in the three species. Gradients are color coded by fMRI networks. DMN, default mode network; Visual, visual network; Auditory, auditory network; Somatomotor, somatomotor network. Adapted with permission from[86] © The Authors, some rights reserved; exclusive licensee AAAS. Distributed under a CC BY-NC 4.0 License (http://creativecommons.org/licenses/by-nc/4.0/"). Adapted with permission from[162].

suggest that resting fMRI activity emerges from instances of segregation between networks interspersed with transient states of temporally overlapping network interactions[176–178]. These dynamic states can be effectively captured by spatially clustering individual fMRI frames into recurring coactivation patterns (CAPs)[177,179]. or, alternatively, by mapping converging spatio-temporal patterns of fMRI activity termed Quasi-Periodic Patterns—QPPs[180]. Both these methods do not rely on second-order statistics (correlations), nor the arbitrary choice of a sliding-window, in addition to maintaining the highest temporal resolution allowed by the sampling of rsfMRI data[172]. These features are especially attractive for cross-species fMRI mapping as they allow for a direct comparison of dynamic fMRI state topography unconstrained by predetermined parcellation schemes.

Whole-brain CAP mapping, first described in humans[179] and then further expanded in a mouse study[181], represent a powerful framework to probe the organization of fMRI dynamics across species. Initial investigations of CAPs as the basis for fMRI dynamics have shown that time-varying fMRI connectivity is governed by a limited number of co-fluctuating network configurations, whose occurrence accounts for more than 60% of the variance in fMRI timeseries[181]. As CAPs capture peaks and throughs of recurring spatiotemporal patterns of BOLD activity, spatially opposed pairs of CAPs can be merged to describe coactivation modes (C-modes), i.e. recurring spatiotemporal patterns of infraslow BOLD activity[59]. Crucially, the topography and dynamic organization of C-modes encompass key evolutionarily relevant features that appear to be conserved across the phylogenetic tree[59]. The most prominent of these, is the topographic organization of a dominant coactivation mode of BOLD activity extending from somatomotor areas to midline components of the DMN, hence spatially reconstituting the principal unimodal-transmodal cortical gradient mapped with static fMRI connectivity (Fig. 4a). Similar patterns of fMRI BOLD activity have been described using QPPs and complex principal component analysis in humans[182]. These findings are of great importance, as they suggest that the microarchitectural organization of cortical areas critically shapes both the static and the dynamic fMRI connectome, a notion that has been formally

explored more in detail in recent rodent[86] and human studies[182]. This observation is also of interest in the light of the natural emergence of functional anticorrelation between somatomotor and DMN systems, a finding that has been observed in multiple species[2,100,176]. Consistent with the emergence of these topographies as a species-invariant phenomenon, the occurrence of C-modes in humans, NHP and mice is time-locked to the infraslow cycle of global fMRI signal[59] (Fig. 4a, inset), and encompass network topographies that can be tracked across evolution in multiple species (Fig. 4b)[59,106]. Specifically, the presence of four phylogenetically related principal C-modes in NHP and humans, three of which exhibited plausible topographic correlates also in the mouse brain was recently demonstrated[59]. Figure 4b shows the prevalent C-mode in humans and its correspondent network topographies in macaques and mice. Beyond the C-mode framework, phylogenetically-relevant correspondences have also been described in anesthetized mice and macaques using region-of-interest approaches[108], or via the detection of QPPs in anesthetized rats, mice, and awake humans[180,183,184]. Collectively, these features define an evolutionarily conserved set of dynamic rules governing fMRI network dynamics and offer opportunities to map and compare evolutionarily relevant dynamic patterns of fMRI activity across the phylogenetic tree.

**Relating fMRI connectivity findings across species.** The identification of evolutionarily conserved principles underlying the organization of fMRI connectivity in the mammalian brain has prompted research aimed at comparing and relating fMRI network organization across the phylogenetic tree. This endeavor is part of the broader field of comparative neuroscience, an emergent area of investigation devoted to understanding how variation in the organization of different brains has unfolded across evolutionary history[185,186]. Attempts to relate fMRI connectivity across species are relatively recent and involve translation strategies characterized by different degrees of complexity. Univariate methods such as region-of-interest approaches (Fig. 5a) have been used to quantify and relate measurements of fMRI connectivity in pre-selected brain regions that are assumed to be homologous across species. This approach has been used to

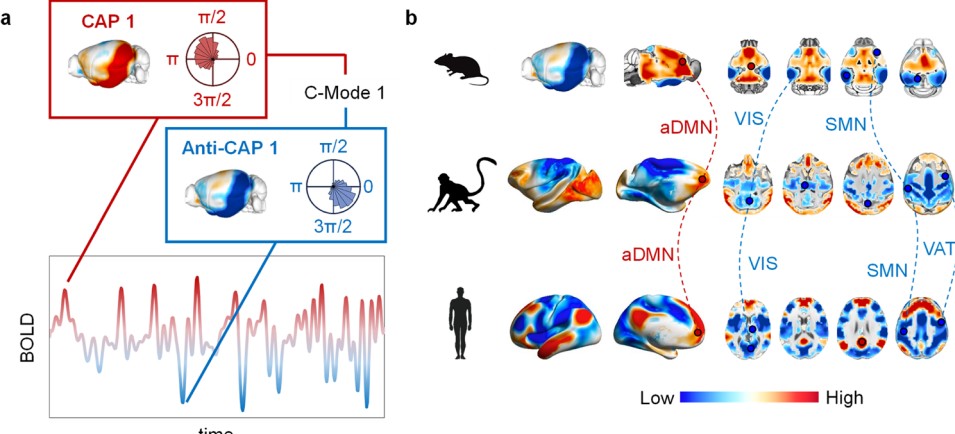

**Fig. 4 fMRI coactivation dynamics is evolutionarily conserved. a** Resting state fMRI activity is dominated by recurring network configurations whose time-varying peaks and troughs can be effectively mapped using frame-wise clustering to produce CAP and anti-CAP pairs, respectively. Red and blue coloring represent peaks and troughs of coherent whole-brain BOLD signals, respectively. Collectively, CAPs and anti-CAPs define a set of evolutionary conserved coactivation modes (C-Modes) whose occurrence is linked to global fMRI signal phases (circular distributions in insets). **b** The spatial topography of C-Modes is largely conserved in awake humans, NHPs and rodents. Red and blue coloring on the mouse brains represent high and low values of BOLD signal, respectively (modified from[59]). Dashed lines denote the evolutionary trajectory of a network's co-activation profile across species in the specified C-mode. aDMN anterior default mode network, pDMN posterior default mode network, SMN somatomotor network, VIS visual network, VAT ventral attention network.

compare network organization and relate regional changes in fMRI connectivity across species both in healthy states[37,117,118,187], as well as in models of brain disorders[71,73,74]. In these studies, regional homology has been typically established by selecting functionally related predefined regions of interest using parcellated anatomical atlases. Although useful in comparing network or connectivity attributes that can be quantified at the regional level, this simple approach fails to capture the multivariate nature of fMRI connectivity and as such it is not optimally suited to highlight the relationship between areas that underlie these interactions.

At the network level, region of interest approaches can be extended to map fMRI connectivity across multiple regions belonging to the same network, or that are centered around a specific hub region, with the aim to identify and compare fMRI network fingerprints[186] (Fig. 5b). A key advantage of such multivariate methods is that they allow for a comparison of whole-networks, and their corresponding connectivity profiles (i.e. interaction with regions within and outside the network) in a semi-quantitative way. This in turn may lead to a precise estimation of the evolutionary trajectory of specific network systems across the evolutionary landscape[125,185,186]. Importantly, the obtained functional interactions can then be related to their underlying anatomical connectivity using white matter fiber bundles[188] or axonal wiring[125], an approach that is particularly powerful in mice where the mesoscale connectome is available[85] and can be directly compared to corresponding whole brain rsfMRI measurements[86]. An interesting extension of this approach is to propose novel functional homologies based on cross-species similarity of inter-regional functional connectivity fingerprints. For example, this method has been recently used to identify a cortical region homolog to macaque area F6 in the human brain, under the assumption that the best candidate for homology would be the human region exhibiting the functional connectivity fingerprint most similar to that observed in the macaque brain[186]. Interestingly, network level approaches can be

used to carry out interspecies comparisons using sensory-driven fMRI responses. By using naturalistic visual stimulation, multivariate methods revealed topologically related convergences extending beyond the primary visual cortex between humans and macaque monkeys[189], and humans and marmosets[190].

Moreover, this approach can also encompass multivariate metrics of functional connectivity beyond Pearson's correlation, thus extending this use to more complex measures of regional interaction and communication[191]. Overall, network-based multivariate methods appear to be optimally suited to compare the topographic organization of specific networks across the phylogenetic tree, with the possible pitfall of requiring the identification and preselection of anchor regions that need to be directly relatable in evolutionarily terms across species. This may not be easily achievable for regions (such as the neocortex) that have greatly expanded and have undergone major reorganization during evolution.

To overcome this issue and allow functional matching across species beyond evolutionarily conserved brain regions, recent years have seen the emergence of a set of complementary methods enabling the quantification of functional connectivity in homologous regions between humans and NHPs, even when their location is decoupled from brain anatomy. These approaches rely on the mono- or multimodal fusion of brain data to create a joint coordinate space, enabling a spatially continuous remapping of evolutionarily relevant anatomical or functional features across the phylogenetic tree. For example, by realigning cortical connectivity gradients of humans and macaque monkeys, joint embedding has been used to capture the common mesoscale brain functional architecture of the two species[162]. Evolutionarily conserved intrinsic coordinate systems can be also built with non-functional data and/or at other scales of investigation. In a recent study[153], spectral embedding was used to align the cortices of pairs of closely related mammalian species, by creating an average feature space that retains shared geometrical and anatomical properties for each pair of species. The iteration of this approach

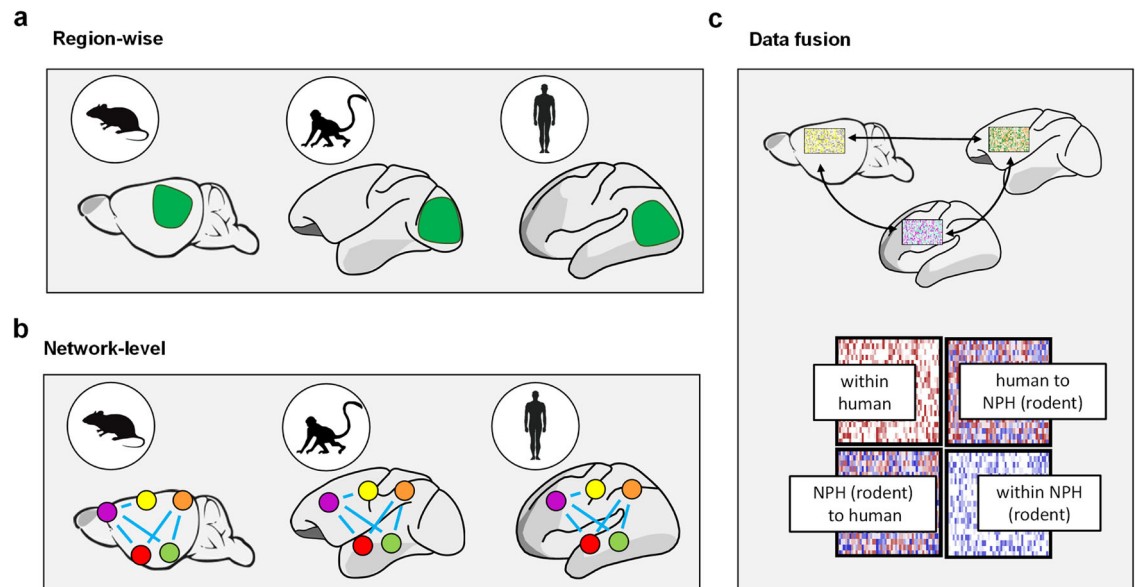

**Fig. 5 Methods to relate fMRI connectivity across species. a** Region-wise approaches are used to quantify rsfMRI connectivity across evolutionarily conserved brain regions. Green indicates the approximate location of visual cortex in mouse, NHP and human brain. **b** Network-level approaches can be used to relate whole fMRI network patterns of connectivity across species via fingerprints encompassing an extended set of evolutionarily relevant anchor points[125]. **c** Fusion-based approaches extend these methods to align fMRI connectivity between species (here referred to NPH, or rodents) using joint embedding of fMRI connectivity per se, or complementary measurements that covary with brain anatomy and connectivity. For example, functional connectivity can be measured in brain regions showing similar profiles of gene expression across species. A major advantage of this approach is the possibility of interpolating the whole cortical space via in between a set of evolutionarily conserved anchor regions.

over 90 brains of extant species enabled a reconstruction of the evolutionary distribution of primary sensory areas across the phylogenetic tree, ranging from ancestral mammals up to non-human primates. Fusion spaces like the one described by Schwartz and colleagues[153] could in principle be used to map and compare fMRI connectivity networks within a unified coordinate system. Finally, leveraging the striking conservation of genetic architecture across mammals, recent attempts to relate brainwide transcriptomic data across species have been described. In one of such studies, regional transcriptomic landmarks in the mouse have been linked to corresponding measures in humans, thus providing a spatial index of anatomical homology between these two species[192]. This research revealed that, as expected, subcortical districts (e.g. caudoputamen) and sensorimotor subdivisions of the neocortex exhibit greater similarity between species compared with polymodal subdivisions of the cerebral cortex. Future extensions of fusion approaches to include multimodal datasets[193] and fMRI connectivity-based metrics (such as gradients, or dynamic topographies) will offer further opportunities to relate anatomical and functional landmarks across species with increased precision (Fig. 5c).

## Challenges of cross-species fMRI

In the preceding sections we have emphasized resemblances in the organization and dynamics of fMRI connectivity networks in rodent, NHPs and humans. Noteworthy evolutionary convergence exists in network organization across species, offering opportunities to extrapolate the results of mechanistic research along the phylogenetic tree, and narrow the "explanatory gap" that exists between microscale modeling and systems levels investigation of brain activity in humans (Fig. 1).

**Moving along the phylogenetic tree**. While increasingly sophisticated approaches are now available to map and compare fMRI connectivity networks and related mechanistic findings across species, a few methodological and conceptual caveats need to be carefully accounted for when extrapolating findings along the evolutionary axis. A first hurdle on the path of a reliable translation of network findings across species, is our fragmentary understanding of how cortical regions have evolved along the phylogenetic tree. This knowledge gap complicates the comparison of functional connectivity results that rely on the use of brain regions that are assumed to be functionally homologous across species. This problem is exemplified by the long-standing debate regarding the existence of a "prefrontal cortex" in rodents[194,195]. Supporting the presence of a prototypical medial prefrontal cortex in rats and mice, plausible cytoarchitectural convergence in the organization of the anterior cingulate have been described in rodents and humans[112]. This finding is consistent with the neuroanatomical organization of the rodent DMN, which includes the whole cingulate cortex as rostral hub[60]. However, the functional relevance of these regions and their role as bona fide substrates of the supramodal cognitive functions typically associated with the (medial) prefrontal cortex in humans remain debated[196]. These controversies underscore the need of exercising caution when specific neocortical locations, and in particular evolutionarily recent polymodal areas, are used as specific anatomical anchor-points for comparing fMRI connectivity networks across species.

Cross-species mapping based on qualitative approaches that are minimally dependent on discrete brain parcellations represents an attractive extension of region-based approaches. One example is to compare whole network systems, e.g. the entire DMN, as opposed to the sole medial prefrontal cortex, or the somatomotor network as opposed to specific motor-sensory sub-domains. The assumption here is that whole fMRI networks can embody more reliably cognate cognitive/behavioral functions than specific brain regions of unclear evolutionarily relevance. The efficacy of this simple approach has been recently demonstrated in neuropsychiatry research, where network-based inferences have facilitated mouse to human extrapolation of connectivity alterations of relevance to autism[71,73,74].

Other methods have expanded this strategy to enable comparisons of RSN organization across species, while minimizing potential bias related to pre-imposed neuroanatomical or network constraints. A notable example of this approach is the recent work from Xu and colleagues[162] in which cross-species matching of fMRI connectivity gradients has been used to project macaque network systems onto human cortical surface. Leveraging evolutionarily conserved hierarchical organization of cortical connectivity (Fig. 3), this work has documented the possibility of improving alignment and interpretation of imaging datasets across species through the use of common coordinate systems[162]. Extensions of this approach down the phylogenetic tree to include rodents are foreseeable, and could be facilitated by the incorporation of intermediate species characterized by lissencephalic neocortical architecture such as marmoset monkeys[140].

Perhaps the most obvious, yet pervasive, caveat to consider when relating fMRI connectivity attributes across the evolutionarily timelines is the fact that higher order cortical functions in human and NHPs have undergone a conspicuous volumetric expansion during evolution. For these reasons, many of these functions (e.g. vocal communication, problem solving) are less specialized or underdeveloped in lower mammalian species, or are subserved by broad cortical substrates that are less specialized or just absent in NHPs and rodents[103,197]. This problem follows the general observation that size and complexity of brain regions are not merely proportional to the variation of total brain volume, area, and cortical folding across evolution[198]. Subcortical systems are somewhat less affected by this issue, as the organizational principles and neural wiring underlying these regions are phylogenetically older, and core to a number of lower-level behavioral domains and physiological responses that appear to be more strongly preserved across evolution. Nonetheless, examples of non-marginal differences in the organization of these systems across species have been reported. For one, basal ganglia connectivity networks exhibit only partly conserved connectivity profiles in rodents and NHPs[125]. Further comparative investigations of subcortical systems are required to provide a proper phylogenetic contextualization of fMRI connectivity topography of non-cortical systems across species.

**fMRI in rodents and non-human primates: is anesthesia the lesser evil?** A methodological consideration of key importance when comparing or extrapolating fMRI connectivity changes across species is the widespread use of light anesthesia and sedation in rodent and NHP imaging. The use of anesthesia follows the need to minimize stress responses and head motion artifacts in fMRI scanning of non-compliant animal species[199,200]. The convenience and ease of use of anesthetized imaging along with the possibility of obtaining long-time series virtually devoid of motion artefacts have greatly contributed to the recent expansions of animal fMRI over the last decade[60,201]. By contrast, human imaging of fMRI connectivity mapping is carried out in awake resting conditions, without the use of sedatives or pharmacological agents.

These procedural discrepancies come with two potential problems: first, differences in fMRI network organization between animals and humans may be compounded by state-related

changes related to the use of anesthesia in animals. A second possible issue pertains to the use of multiple anesthetic and sedative combinations in different species, each of which may bias the functional organization of fMRI network in different ways (reviewed in[202]). As a result, fMRI network comparisons between animal species may suffer from anesthetic-related bias.

While these issues complicate inferences that can be made at the level of networks across species, retrospective analyses of the fMRI literature suggests that their impact can be mitigated by choice of anesthetic regimes that preserve network organization[100,199,203]. Specifically, converging evidence suggests that several sedative regimens in rodents and NHP broadly preserve the functional organization of RSNs with respect to the structure of underlying brain anatomy[37,86,106,135,204]. Analogous observations have been reported in humans, where differences in awake versus sedated states produced by medetomidine appeared to be small, and very focal compared to the broad extension of brain networks in awake and sedated state[205]. Moreover, the spatial association between functional and axonal connectivity in anesthetized rodents and NHPs is robust[86,131], recapitulating the general good agreement between these quantities observed in awake humans[4].

These observations suggest that the use of anesthesia per se does not prevent meaningful comparisons of network features across species, provided the anesthetic protocol is judiciously chosen and validated with reference to the underlying anatomical structure of brain networks. fMRI imaging under light sedation is thus a practice that is commonly used in the preclinical fMRI community[71,74,89,111], as it combines readiness of use with the potential of recording long time-series with negligible motion artifacts.

Importantly, the possibility of obtaining robust fMRI connectivity mapping with sedation in animals does not imply that pharmacological agents negligibly affect the organization of brain connectivity, or its underlying neurovascular coupling. Important interactions between anesthesia and hemodynamic signals have indeed been reported[206–209] which may confound the interpretation and topological organization of fMRI networks. Moreover, different anesthetic mixtures can result in different patterns on connectivity[100,210], and may exert different effects across species[208]. Owing to these differences, it is important that mechanistic studies, whenever possible, report the generalizability of findings across anesthetic regimens, or their extension to awake conditions[16,106]. Of equal importance, the administration of mixtures of anesthetic agents, a procedure frequently employed in both rodent and NHP studies[203], can present latent interpretational and technical difficulties related to the stability of the ensuing neurophysiological parameters in multimodal investigations[16]. The main problem associated with the use of anesthetic cocktails is the presence of major physiological drifts in neural rhythms and brain states reflecting the pharmacokinetic properties of the employed drugs and ensuing receptor desensitization mechanisms[211]. This effect is often not patently detectable in terms of global network configuration but becomes apparent when fMRI is coupled with more sensitive measures of neural activity (such as electrophysiological recordings). In this respect, the application of single-anesthetic preparations[200] may offer advantages over drug cocktails in terms of brain state stability.

To circumvent these problems, recent years have seen a major drive towards the implementation of fMRI connectivity mapping in awake NHP[53,212] and rodents[102,106,213]. Most awake fMRI protocols in animals deal with the problem of head-related motion and restraint-induced stress via the use of progressive habituation to scanner environment and the use of customized helmet[214], head or body-restraint devices[106,215,216]. In NHPs,

these strategies have been recently combined with the use of naturalistic viewing paradigms, with very encouraging results in terms of reduced head motion, and the possibility of harmonizing states between human and NHP fMRI data[189,190,217]. These approaches have shown the possibility of reliably mapping fMRI networks with minimal contamination by motion and stress, paving the way for a broader use of awake rsfMRI mapping in the preclinical field. Not surprisingly, head-to-head comparisons of awake and anesthetized animals have revealed subtle but important differences in the organization of fMRI networks. Specifically, while the overall structure of RSN networks is broadly comparable between anesthesia and awake states, focal changes in fMRI network organizations have been consistently reported in rodents and primates[53,106,218]. These include a stronger involvement of arousal-related forebrain nuclei, an increased anti-correlation between network sub-systems, and a weakening of structure-function coupling in awake states such to maximize between-network communication[53,106,218]. Moreover, time-varying fMRI connectivity presents stereotypical trajectories that are predictive of the consciousness in rodents, NHP, and humans[53,106,218–220]. These are important factors that can aid the interpretation of results from investigations regarding rsfMRI network dynamics in humans and animal models.

As promising as awake animal imaging is, it also comes with some notable limitations. For example, head-fixation usually involves invasive surgeries that may result in susceptibility artefacts that may degrade image quality. Similarly, habituation to the scanner is typically achieved through a lengthy and progressive exposure to restraint, hence making the whole procedure long and labor intensive[106,221]. Awake imaging is also predictably associated with higher levels of head motion, hence potentially polluting the quality of resulting images. More importantly, even though corticosterone levels in head-fixed, scanner acclimated rodents suggest extensive habituation results in marginal stress levels[106], questions remain as to whether the "resting-state" that is attainable upon scanner acclimation does actually recapitulate the quiet wakefulness that characterizes human fMRI experiments. In this respect, the differences observed between static fMRI network organization in awake and anesthetized humans[205] seem to be qualitatively more focal and smaller than those found between awake and anesthetized rodents or NHP[53,106], suggesting that this might not be the case. Thereby, the possibility that awake animal imaging entails considerably higher arousal conditions than human fMRI mapping should be given serious consideration and needs to be accounted for when comparing network topography across species. fMRI mapping in high arousal-related states could be especially noxious in brain stimulation studies, where stimulus-locked motion can occur in response to stimulus-unrelated arousing cues (such as light detection in optogenetics, or interoceptive signals), hence contaminating fMRI timeseries. Because of these issues, the use of anesthesia is a practice that still has (and will retain) value in the animal fMRI community, where it is regarded by some as lesser (methodological) evil, owing to its readiness of use and the high quality of the data obtainable. However, despite its growing pains, awake fMRI in animals will soon reach methodological maturity, and will thus soon steadily complement the use of anesthesia in animal fMRI.

## Concluding remarks

Advancements in the implementation of fMRI connectivity mapping across species offer a privileged angle of investigation to probe the evolutionary trajectory or the mammalian brain at the network level, and to mechanistically decode the elusive physiological mechanisms governing large scale network organization.

The recent inception of methods enabling reliable fMRI network mapping in rodents is a welcome addition to the methodological repertoire available to neuroscientists interested in these research themes.

Form a comparative neuroscience standpoint, major correspondences have been identified in the large-scale organization of fMRI networks across the mammalian realm, revealing phylogenetically conserved network systems (e.g. salience, default, motor-sensory etc.), a dominant axis of cortical fMRI connectivity, and topographically-consistent dominant fMRI substates. These general principles are overlaid onto species-specific systems that are evolutionarily more recent, and that as such cannot be straightforwardly translated along the phylogenetic axis. Notwithstanding these evolutionary constraints, and the inevitable growing pains of this relatively new field of comparative neuroscience, the future of cross-species fMRI is bright, and poised to greatly advance our understanding of the physiological basis and organizational principles of fMRI connectivity in health and disease.

**Reporting summary**. Further information on research design is available in the Nature Portfolio Reporting Summary linked to this article.

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

## Acknowledgements

This work has received funding from the European Research Council (ERC) under the European Union's Horizon 2020 research and innovation program (#DISCONN; no 802371 to AG). AG also acknowledges funding from the Telethon foundation (GGP19177), Simons Foundation (SFARI 982347). MP is supported by European Union's Horizon 2020 research and innovation programme (Marie Sklodowska-Curie Global Fellowship - CANSAS, GA845065). EDG is supported by the Canadian Institute of Health Research (Postdoctoral fellowship—MFE187902). TX is supported by Brain Initiative National Institutes of Health (RF1MH128696).

## Author contributions

We confirm contribution to the paper as follows: study conception: A.G.; literature review: M.P., D.G.-B., E.D.G., T.X., A.G. draft manuscript preparation: M.P., D.G.B., E.D.G., A.G. All authors reviewed the results and approved the final version of the manuscript.

## Competing interests

The authors declare no competing interests.
