## [Peer Review File · Communications Biology]

Reviewers' comments:

Reviewer #1 (Remarks to the Author):

This review explores the current challenges and new developments in the field of fMRI as it is applied to understanding functional connectivity in humans and model organisms (specifically rodents and NHPs). The authors do an excellent job of situating this field within the broader comparative neuroscience/neuroimaging research, identifying challenges to validating the functional and anatomical correlates of these networks within species as well as across species, especially given that different species have different methodological toolkits available for them.

This review does a commendable job of surveying recent multidisciplinary lines of evidence correlating function and structure with these networks, as well as summarizing the new approaches that are being developed to accurately compare these networks across species.

The discussion on limitations and important considerations on the use of anesthesia in fMRI, as well as the discussions on evolutionary changes to mammalian brains across taxa, shows a deep knowledge of these areas.

I don't have any suggestions for modifications, the paper is very well-written and provides a very useful survey of the present and future use of fMRI in understanding network activations in an evolutionary context. I recommend Accept.

Reviewer #2 (Remarks to the Author):

Pagani and colleagues summarized the research about fMRI connectivity networks across species, focusing on fMRI connectivity mapping. Their objective is to better understand the functional connectome's mechanisms, which is crucial for comprehending how various parts of the brain collaborate. One significant discovery is that the organization of mammalian connectivity networks across species allows for comparisons and extrapolations of fMRI data between different types of animals.

The review is well structured and gives an accurate and yet accessible overview of the field. I have a few minor comments:

- 1) p.4 line 88: it is noted "reviewed by" without putting the name of the study. It would be more appropriate to put "reviewed by Suarez et al.4".
- 2) Check that all terms "(Figure x)" are bolded or at least marked in a uniform manner.
- 3) p.4 line 101: the authors refer to Figure 1 after mentioning Figure 1A. Perhaps it would be better to talk about it first.
- 4) p.5 lines 107-110: recent investigations are mentioned but no references are given.
- 5) Figure 1A: the text and the figure are not very clear. Do "Neural Systems" correspond to the human species? I think it should be modified to improve comprehension.
- 6) Figure 1B: it is noted "monkeys" in the figure but in the text and review, the term NHP is preferred. This needs to be harmonized. Moreover, the color of "special circumstances" is not noted in the text of the figure. 1, 2, 3... are not very visible. Revise the color or something else to make it more visible. Finally, some terms are defined in the text but not present in the figure (example: TMS, tACS).
- 7) p.6 line 130: the term "NHP" is only defined here whereas it has been used since Figure 1, it must be done before.
- 8) p.6 line 146: I just wanted to mention that I think it would be helpful to have more than one reference when discussing the impact of anesthetics on fMRI.
- 9) p.6 lines 146-147: the results on transgenic primates should be described a little more fully.
- 10) I noticed that the same paragraph appeared twice in the review (p.7 lines 167-177 and p.7-8 lines 198-208).

- 11) p.9 line 247: (i.e.: marmosets, lemur monkeys...)
- 12) Figure 2A: it is necessary to increase the size of the figure to improve visibility and understanding.
- 13) Figure 2A and 2B: Harmonizing the colors of the networks on these two figures would make them easier to read and understand.
- 14) Figure 3A, 3B and 3C: (page 13, lines 370-375), the related text is really succinct, more information is needed. In the Figure 3B, color legend must be added. In the Figure 3C, the term "auditory" is not represented in monkeys and human. Is it normal or not? In addition, the mouse part needs to be moved away from the macaque part, as there is a slight overlap between the captions.
- 15) p.6 line 131: the term of BOLD is not defined.
- 16) p.14 line 411; p.15 lines 440-442; p.17 lines 488-489, p.18 line 517: references are not in the same format as the rest of the review.
- 17) Figure 4B: I think it would be helpful to increase the size of the figure to improve legibility.
- 18) p.15 line 435: there is no Figure 4C.
- 19) Figure 3B: why are two colors used to designate the thalamus? It is not understandable, the authors will have to explain in the text.
- 20) p.18 lines 524 and 525: references are not noted at the end of the sentence.
- 21) Figure 5C: I don't understand why macaques is not really represented on the lower part of the figure.
- 22) p.21: A summary table of the effects of anesthetics on fMRI across species would be useful.
- 23) p.23 line 665: "\\\" is present, I think it is a typing error.

We would like to thank the Editor and Reviewers for their positive evaluations, and constructive comments, and for the opportunity to submit a revised manuscript. We believe the resulting manuscript is improved following their suggestions. Please find below a point-by-point response to the reviewers' comments (our response in blue typeface). We have also added below the text edits we have introduced in the manuscript in response to the reviewers' comments (yellow highlight).

Reviewers' comments:

Reviewer #1 (Remarks to the Author):

This review explores the current challenges and new developments in the field of fMRI as it is applied to understanding functional connectivity in humans and model organisms (specifically rodents and NHPs). The authors do an excellent job of situating this field within the broader comparative neuroscience/neuroimaging research, identifying challenges to validating the functional and anatomical correlates of these networks within species as well as across species, especially given that different species have different methodological toolkits available for them.

This review does a commendable job of surveying recent multidisciplinary lines of evidence correlating function and structure with these networks, as well as summarizing the new approaches that are being developed to accurately compare these networks across species.

The discussion on limitations and important considerations on the use of anesthesia in fMRI, as well as the discussions on evolutionary changes to mammalian brains across taxa, shows a deep knowledge of these areas.

I don't have any suggestions for modifications, the paper is very well-written and provides a very useful survey of the present and future use of fMRI in understanding network activations in an evolutionary context. I recommend Accept.

We thank the reviewer for the words of appreciation.

Reviewer #2 (Remarks to the Author):

Pagani and colleagues summarized the research about fMRI connectivity networks across species, focusing on fMRI connectivity mapping. Their objective is to better understand the functional connectome's mechanisms, which is crucial for comprehending how various parts of the brain collaborate. One significant discovery is that the organization of mammalian connectivity networks across species allows for comparisons and extrapolations of fMRI data between different types of animals. The review is well-structured and gives an accurate and yet accessible overview of the field. I have a few minor comments:

1) p.4 line 88: it is noted “reviewed by” without putting the name of the study. It would be more appropriate to put “reviewed by Suarez et al.⁴”.

We thank the reviewer for pointing out this omission. We have reworded it as follows: “... concurrently with rsfMRI (reviewed by Suárez et al.,¹) Similarly, large-scale imaging initiatives ...”.

2) Check that all terms “(Figure x)” are bolded or at least marked in a uniform manner.

We have made all the items “(Figure x)” in bold in the revised manuscript.

3) p.4 line 101: the authors refer to Figure 1 after mentioning Figure 1A. Perhaps it would be better to talk about it first.

We thank the reviewer for spotting this typo. Here, we meant to refer to Figure 1A, not Figure 1. We have reworded as follows: “... to bridge the “explanatory gap” (Figure 1A). Specifically, the combination of rsfMRI ...”

4) p.5 lines 107-110: recent investigations are mentioned but no references are given.

We here referred to the studies we describe in great detail in the section that follows. So, there is no need to emphasize studies that are discussed at length elsewhere in our review.

The whole sentence now reads as follows: “The impact and potential of mechanistic investigation in animals is epitomized by a series of investigations that we briefly summarize in the following sections, with the intent to make readers aware of the critical contribution and impact preclinical fMRI is having in the field”.

5) Figure 1A: the text and the figure are not very clear. Do “Neural Systems” correspond to the human species? I think it should be modified to improve comprehension.

We thank the reviewer for pointing this out. We have edited Figure 1A to make clear that “Neural Systems” in bottom left panel refers to Neural Systems in humans. We have also reworded the text and the caption of Figure 1 as following:

- “... clinical scores and rsfMRI network dysfunction in humans, hence substantiating the role of distributed network ...”.
- “... gap exists between models of human brain function at different levels of ...”.

6) Figure 1B: it is noted “monkeys” in the figure but in the text and review, the term NHP is preferred. This needs to be harmonized. Moreover, the color of “special circumstances” is not noted in the text of the figure. 1, 2, 3... are not very visible. Revise the color or something else to

make it more visible. Finally, some terms are defined in the text but not present in the figure (example: TMS, tACS).

We thank the reviewer for pointing out these inconsistencies. We have used the term “NHP” instead of “monkey” in the revised manuscript and in Figure 1. We have also reworded the corresponding text as follows:

- “ ... We note however that reports of a cortical system subserving executive control functions in NHP have been published ... ”
- “ ... salience, limbic, somato-motor and visual networks in the human, NHP and mouse brain ...”

The colour of “special circumstances” is yellow, we have added it to the revised manuscript. To improve visibility, we have increased the font size and used black colour for the numbers 1 to 7. Finally, we have now provided a description for all the abbreviations in the caption as follows:

- ... Non-invasive stimulation includes transcranial magnetic stimulation (TMS), transcranial alternating current stimulation (tACS) and related methods ...

7) p.6 line 130: the term “NHP” is only defined here whereas it has been used since Figure 1, it must be done before.

We thank the reviewer for pointing out this inconsistency. We have now defined the term NHP the very first time it is used: “ ... can be implemented in humans, non-human primates (NHPs) and rodents to ...”

8) p.6 line 146: I just wanted to mention that I think it would be helpful to have more than one reference when discussing the impact of anaesthetics on fMRI.

We have now expanded our discussion on fMRI in anesthetized and awake monkeys as follow:

“... The use sedative and anaesthetic agents has also been explored to map and compare patterns of fMRI activity as a function of brain state. For example, a study comparing wakeful versus unconscious states has shown that spontaneous functional connectivity patterns in awake monkeys show a rich repertoire of functional connections that is more dissimilar to structural connectivity compared to anaesthesia^{2, 3}. Differences in cross-subjects variability of fMRI connectivity patterns has been described in the cortex of awake vs. anesthetized NHP⁴. Cortical BOLD responses to visual stimulation are instead consistent in awake vs. unconscious NHP⁵. Finally, it has also been shown that the intrinsic network structure of main primary and associative networks in the anesthetized macaque monkeys are topographically similar to those mapped in conscious humans⁶.”

9) p.6 lines 146-147: the results on transgenic primates should be described a little more fully.

We have now expanded our discussion of fMRI in TRANSGENIC PRIMATES as follows:

“... Finally, notable attempts to map fMRI connectivity in transgenic primate models of human disorders have also been recently described ⁷. These include evidence that MECP2 overexpression in macaque monkey results in hyperconnectivity of prefronto-cingulate networks ⁸, or the identification of local and global fMRI connectivity alterations in transgenic macaques with autism-associated SHANK3 mutation ⁹.

10) I noticed that the same paragraph appeared twice in the review (p.7 lines 167-177 and p.7-8 lines 198-208).

We thank the reviewer for pointing this out. We have removed the duplication.

11) p.9 line 247: (i.e.: marmosets, lemur monkeys...)

We have now fixed this typo.

12) Figure 2A: it is necessary to increase the size of the figure to improve visibility and understanding.

We thank the reviewer for pointing this out. We have increased the font size of Figure 2A.

13) Figure 2A and 2B: Harmonizing the colors of the networks on these two figures would make them easier to read and understand.

We have now harmonized the colours of panel 2A and 2B. The resulting figure is now easier to understand.

14) Figure 3A, 3B and 3C: (page 13, lines 370-375), the related text is really succinct, more information is needed. In the Figure 3B, color legend must be added. In the Figure 3C, the term “auditory” is not represented in monkeys and human. Is it normal or not? In addition, the mouse part needs to be moved away from the macaque part, as there is a slight overlap between the captions.

We thank the reviewer for pointing this out. We have expanded that section as following:

“... Collectively, these lines of evidence suggest that the spatial topology of cortical fMRI connectivity along preordered evolutionarily relevant gradients is a fundamental organizational principle of mammalian fMRI connectivity across species. As shown in Figure 3A, gradients of functional connectivity in the human cortex are arranged along two main dimensions. One end

of the principal gradient includes unimodal brain regions such as visual, motor and somatosensory areas. The other end includes polymodal cortical regions, such as those included within the default mode network. A second gradient spans the sensory-specific arrangement of cortical regions, from visual to somatomotor areas. Importantly, the organization of these two principal gradients is evolutionarily-conserved in lower mammal species such as the mouse and NHPs (Figure 3B). The topological organization of the two principal gradients segregates the default mode network vs primary rsfMRI networks, and modality-specific rsfMRI networks in mice, NHP and humans (Figure 3C). Thereby, above and beyond the use of network-specific inferences, this organizational axis may provide valuable ...”.

We have also added the colour legend to Figure 3B to describe distribution of polymodal-unimodal cortical gradient as previously done by others. We have indicated the auditory cortex in Figure 3C as that is an important hub of the mouse fMRI gradients¹⁰. As discussed in our previous work, the fact this area is located at the far end of the unimodal-gradient in rodents but not in humans probably reflects a lower functional specialization of the cortical mantle in rodents compared to higher mammalian species. Finally, we have also moved away the plot of the mouse to remove the overlap between captions.

15) p.6 line 131: the term of BOLD is not defined.

We thank the reviewer for pointing this out. We have reworded the text as follows: “... understanding of the neural correlates of Blood Oxygenation Level Dependent (BOLD) fMRI activity ...”

16) p.14 line 411; p.15 lines 440-442; p.17 lines 488-489, p.18 line 517: references are not in the same format as the rest of the review.

We thank the reviewer for pointing out these inconsistencies. We have corrected the format of those references.

17) Figure 4B: I think it would be helpful to increase the size of the figure to improve legibility.

We have now increased the font size of Figure 4A-B.

18) p.15 line 435: there is no Figure 4C.

We thank the reviewer for spotting this typo. We have removed the reference to Figure 4C in the text.

19) Figure 3B: why are two colors used to designate the thalamus? It is not understandable, the authors will have to explain in the text.

We thank the reviewer for highlighting this inconsistency. The dotted lines here denoted the cross-species trajectory of network co-activation within neuroanatomically-related C-modes ¹¹. In the case of the thalamus, in humans and macaques this region happens to be co-deactive (i.e. it has negative BOLD polarity), while in mice, it is co-active (it has positive BOLD polarity), with the rest of the pattern of coactivation (termed here C-mode). We agree that emphasizing focal differences in polarity across species in the face of large overlap in the organization of critical polarity may be confusing and could in fact counter the point that us and others have identified similarities in the dynamic organization of fMRI connectivity (think of CAPs, or QPPs). For these reasons, we have now removed reference to thalamic areas in the figure and only highlighted areas exhibiting coherent coactivation profiles.

We agree with the reviewer and have specified this in the text and figure caption as follows:

- “... was recently demonstrated¹¹. Figure 4B shows the prevalent C-mode in humans and its correspondent network topographies in macaques and mice. Beyond the C-mode framework ...”
- “... and rodents. Red and blue coloring on the mouse brains represent high and low values of BOLD signal, respectively (modified from Gutierrez-Barragan et al., 2023). Dashed lines denote the evolutionary trajectory of a network’s co-activation profile across species in the specified C-mode. aDMN, anterior default mode network ...”

20) p.18 lines 524 and 525: references are not noted at the end of the sentence.

We have now added the missing reference in the sentence (Schwartz et. al 2023).

21) Figure 5C: I don’t understand why macaques is not really represented on the lower part of the figure.

We thank the reviewer for pointing out this inconsistency. We have substituted “mouse” with “NPH (rodent)” and updated the corresponding figure caption accordingly.

22) p.21: A summary table of the effects of anesthetics on fMRI across species would be useful.

We thank the reviewer for this suggestion. Large variability exists between the mixtures of anaesthetic used across species and their putative impact on fMRI connectivity. In the lack of comprehensive and rigorous assessment of how different aesthetic may bias fMRI connectivity within and across species, we think putting together a table might result in a speculative exercise that is beyond the scope of the present work. We have however referred readers interested readers to a recent reviews on the subject ¹².

The corresponding paragraph now reads as follows: “These procedural discrepancies come with two potential problems: first, differences in fMRI network organization between animals and humans may be compounded by state-related changes related to the use of anesthesia in animals. A second possible issue pertains to the use of multiple anesthetic and sedative combinations in different species, each of which may bias the functional organization of fMRI network in different ways (reviewed in¹²). As a result, fMRI network comparisons between animal species may suffer from anesthetic-related bias.”

23) p.23 line 665: “\” is present, I think it is a typing error.

We have now removed this typing error.

References

1. Suárez LE, Markello RD, Betzel RF, Misic B. Linking Structure and Function in Macroscale Brain Networks. *Trends in Cognitive Sciences*, (2020).
2. Barttfeld P, Uhrig L, Sitt JD, Sigman M, Jarraya B, Dehaene S. Signature of consciousness in the dynamics of resting-state brain activity. *Proceedings of the National Academy of Sciences* **112**, 887-892 (2015).
3. Hahn G, *et al.* Signature of consciousness in brain-wide synchronization patterns of monkey and human fMRI signals. *NeuroImage* **226**, 117470 (2021).
4. Xu T, *et al.* Interindividual variability of functional connectivity in awake and anesthetized rhesus macaque monkeys. *Biological Psychiatry: Cognitive Neuroscience and Neuroimaging* **4**, 543-553 (2019).
5. Logothetis NK, Guggenberger H, Peled S, Pauls J. Functional imaging of the monkey brain. *Nature neuroscience* **2**, 555-562 (1999).
6. Vincent JL, *et al.* Intrinsic functional architecture in the anaesthetized monkey brain. *Nature* **447**, 83-86 (2007).

7. Huffaker SJ, *et al.* A primate-specific, brain isoform of KCNH2 affects cortical physiology, cognition, neuronal repolarization and risk of schizophrenia. *Nature medicine* **15**, 509-518 (2009).
8. Cai D-C, *et al.* MECP2 duplication causes aberrant GABA pathways, circuits and behaviors in transgenic monkeys: neural mappings to patients with autism. *Journal of Neuroscience* **40**, 3799-3814 (2020).
9. Zhou Y, *et al.* Atypical behaviour and connectivity in SHANK3-mutant macaques. *Nature* **570**, 326-331 (2019).
10. Coletta L, Pagani M, Whitesell JD, Harris JA, Bernhardt B, Gozzi A. Network structure of the mouse brain connectome with voxel resolution. *Science Advances* **6**, eabb7187 (2020).
11. Gutierrez-Barragan D, Panzeri S, Xu T, Gozzi A. Evolutionarily conserved fMRI network dynamics in the human, macaque and mouse brain. *Submitted*, (2023).
12. Mandino F, *et al.* Animal functional magnetic resonance imaging: trends and path toward standardization. *Frontiers in Neuroinformatics* **13**, 78 (2020).

REVIEWERS' COMMENTS:

Reviewer #2 (Remarks to the Author):

I thank the authors for taking my comments into account. I have no further suggestions for changes. I recommend Accept.